# A Lightweight Fusion Distillation Network for Image Deblurring and Deraining [note 1]

**DOI:** 10.3390/s21165312

**Published:** 2021-08-06

**Authors:** Yanni Zhang, Yiming Liu, Qiang Li, Jianzhong Wang, Miao Qi, Hui Sun, Hui Xu, Jun Kong

**Affiliations:** 1College of Information Science and Technology, Northeast Normal University, Changchun 130000, China; zhangyn500@nenu.edu.cn (Y.Z.); liq782@nenu.edu.cn (Q.L.); wangjz019@nenu.edu.cn (J.W.); qim801@nenu.edu.cn (M.Q.); 2Institute for Intelligent Elderly Care, Changchun Humanities and Sciences College, Changchun 130000, China; sunh333@nenu.edu.cn; 3College of Information Science and Engineering, Hunan Normal University, Changsha 410000, China; yimingliu321@163.com

**Keywords:** image deblurring, image deraining, lightweight distillation block, fusion mechanism

## Abstract

Recently, deep learning-based image deblurring and deraining have been well developed. However, most of these methods fail to distill the useful features. What is more, exploiting the detailed image features in a deep learning framework always requires a mass of parameters, which inevitably makes the network suffer from a high computational burden. We propose a lightweight fusion distillation network (LFDN) for image deblurring and deraining to solve the above problems. The proposed LFDN is designed as an encoder–decoder architecture. In the encoding stage, the image feature is reduced to various small-scale spaces for multi-scale information extraction and fusion without much information loss. Then, a feature distillation normalization block is designed at the beginning of the decoding stage, which enables the network to distill and screen valuable channel information of feature maps continuously. Besides, an information fusion strategy between distillation modules and feature channels is also carried out by the attention mechanism. By fusing different information in the proposed approach, our network can achieve state-of-the-art image deblurring and deraining results with a smaller number of parameters and outperform the existing methods in model complexity.

## 1. Introduction

Image deblurring and Fderaining are both typical and essential tasks in image restoration research. Blurry images are caused by the movement of objects or camera shaking, and rainy images often distort the color of the background image due to the blocking and refraction of sunlight by rain. Both of theses affect the vision sensors and people’s visual perception of detailed image information. Therefore, image deblurring and deraining have become indispensable steps for many computer vision tasks, such as object detection, image classification, and surveillance. However, estimating the blur kernel and rain streaks to restore sharp and clean images from the blurry and rainy ones is difficult since it is a highly ill-posed problem. Most traditional methods [1,2,3,4,5,6] regularize the solution space by introducing some prior information to model the blur kernel and rain streaks. However, because the prior information may not conform to the real scene, the quality of the restored images is not optimal. Recently, some information fusion based methods have also been proposed for image restoration problem. For instance, Zhu [7] utilized a set of artificial images to construct pixelwise weight maps, so that both global and local image features can be analyzed and fused for image dehazing. In [8], a multiple image fusion scheme based on adaptive structural decomposition was also proposed for haze image enhancement. While these information fusion methods can avoid prior information estimation and restore the image without inversing the image hazing model, they still belong to the shallow image restoration model. Therefore, the methods in [7,8] rely on carefully designed filter, criterion or measure to characterize the image features.

With the rapid development of deep learning, many convolutional neural network (CNN) based methods [9,10,11,12,13,14,15,16,17,18,19,20,21,22] have been proposed for effective image deblurring [9,10,11,12,13,14,15,16,17,18,23] and deraining [19,22,24,25,26,27,28]. Compared with the traditional shallow models [1,2,3,4,5,6], these deep CNN methods do not need to estimate blur kernel or rain streaks, but directly predict clear images from the degradation ones. Furthermore, they also avoid careful engineering to design image feature extractor. While these methods are effective, they still have two shortcomings. First, most of the existing deep CNN methods cannot distillate the useful features from all extracted features, so some useless feature information will interfere with the network and prevent the model from recovering the details of the image. While some recent studies attempt to solve this problem by employing the attention mechanism [19,23,25,27,29], they still have drawbacks. MPRNet [25] added a supervised attention module (SAM) between every two stages to enable progressive learning. However, only the local image information was focused in SAM. Some deraining [19,27] and deblurring [23,29] methods adopted the channel attention mechanism [30] to filter useful feature. However, the channel attention mechanism [30] just assigned different weights to different feature maps rather than reorganize useful information. The second problem of the existing deblurring and deraining methods is the complexity of the network model. In order to take full advantage of the surrounding pixels to recover image details, multi-scale methods [9,10,11,12,13,25] are commonly adopted in the image deblurring and deraining task. While this multi-scale strategy can obtain different receptive fields at different scales, it sacrifices the complexity of the network. Nah et al. [9] introduced three kinds of blurry images with different sizes into the model, which required the network to carry out feature extraction and image reconstruction three times. Therefore, the number of network parameters is too large. SRN [10] employed the LSTM mechanism to share feature extraction results across scales, but it still failed to solve the problem fundamentally. Zhang et al. [13], DeblurGanv2 [16], PReNet [22], MPRNet [25] also tackled the deraining and deblurring problem in multiple stages, which inevitably leads to the increase of network parameters.

This paper proposes a lightweight fusion distillation network (LFDN) to address the problems mentioned above. Compared with the existing methods, our LFDN could successfully recover the image details from the blurry and rainy images with a lightweight network structure, as shown in Figure 1. The contribution can be summarized as follows:We propose a multi-scale hierarchical information fusion scheme (MSHF) to encode the image with rain and blur. MSHF extracts and fuses the image feature in multiple small-scale spaces, which can eliminate redundant parameters while maintaining the rich image information.We propose a very lightweight module named feature distillation normalization block (FDNB) which can constantly filter out useless feature channel information. To the best of our knowledge, it is the first time that the distillation network is adopted in image deblurring and deraining tasks.Two attention mechanism based modules are also presented in the decoding process of our approach to exploit the interdependency between the layers and feature channels, which is termed as Multi-feature fusion module based on attention mechanism (MFFD). Through MFFD, a better information fusion can be achieved to compensate for the potential image detail lost in FDNB.

## 2. Related Work

Image Deblurring. In recent years, CNNs based image deblurring has been developed rapidly. Nah et al. [9] proposed a multi-scale CNN method for image deblurring called DeepDeblur, which is based on a coarse-to-fine strategy to restore the sharp image progressively. While it achieved satisfactory results, DeepDeblur contains 40 convolutional layers in each scale without parameter sharing. Zhang et al. [13] proposed a spatial variant neural network that consists of three CNNs and a recurrent neural network (RNN) for dynamic scene deblurring. While this algorithm is effective, it uses many convolutional layers for feature extraction and weights estimation by RNNs, which increase the number of parameters. Tao et al. [10] employed an encoder–decoder structure to propose a scale-recurrent network (SRN). Compared with DeepDeblur, SRN applied the long-short term memory (LSTM) to share weights across scales. Similarly, the parameter sharing scheme is also adopted by Gao et al. [11] to improve the efficiency of their image deblurring network. Zhang et al. [12] investigated a new scheme that exploits the deblurring cues at different scales via a hierarchical multi-patch model and proposed a simple yet effective multi-level CNNs model called Deep Multi-Patch Hierarchical Network (DMPHN). Nevertheless, under the coarse-to-fine scheme, most networks use a large number of training parameters due to large filter sizes. Thus, the multi-scale, scale-recurrent, and multi-patch methods result in an expensive runtime and struggle to improve deblurring quality. In order to solve this problem and further make use of multi-scale mechanisms, Kupyn et al. [16] introduced a feature pyramid network (FPN) [31] to replace the multi-scale strategy, which can handle both semantic information and context detail of the burry image. However, the problem of semantic information dilution occurs in its decoding process.

Image Deraining. Deng et al. [18] constructed a specialized detail repair network that utilizes some well-designed structure detail context aggregation blocks (SDCAB) for image deraining. However, SDCAB only focused on local feature fusion, which may lead its result to be suboptimal. Jiang et al. [19] employed a recurrent manner to capture the global image texture, thus allowing to explore the complementary and redundant information at the spatial dimension to characterize target rain streaks. However, this method required both rainy image and rain strike information as the input of network. Thus, it cannot be applied to most benchmark datasets which only contain rainy images without the rain strike information. Recently, an uncertainty guided multi-scale residual learning network (UMRL) [28] was proposed to learn the rain pattern information at different scales. At the same time, a cycle spinning frame was also used to remove artifacts. While UMRL achieved efficient results, it needs three kinds of images at different scales as input, which inevitably increases the model complexity. Ren et al. [22] provided a better and simpler baseline deraining network by simultaneously considering network architecture, input, output, and loss functions. MPRNet [25] proposed a multi-stage architecture that progressively learns restoration functions for the degraded inputs. Specifically, MPRNet adopted the encoder–decoder architecture to learn context-specific features and then combined them with high-resolution branching to retain local information. However, MPRNet ignored the global information. RESCAN [27] defined heavy rain as the accumulation of multiple rainwater layers. Because rainwater stripe layers overlap each other, it is not easy to remove rainwater completely in a single stage. Therefore, they proposed a deep network combined with a recursive neural network that preserves useful information in the first stage and then facilitates the removal of rainwater in the second stage. However, the multi-stage strategy makes RESCAN suffer from the problem of model complexity.

Attention Mechanism. Inspired by its successful applications in natural language processing, the attention mechanism has been widely used in image processing tasks [19,25,29,32,33,34,35]. Zhang et al. [32] leveraged the attention mechanism to allow the network to focus on the relationship among spatial image areas. Kuldeep et al. [29] utilized the weighted sum of all location features to selectively aggregate the location feature. Kim et al. [33] learned the correlation between feature channels through residual blocks and spatial channel attention. MSPFN [19] combined pyramid structure [31] and channel attention mechanism [30] to synergistically represent multi-scale rain-pattern information. More recently, Niu et al. [35] proposed a holistic attention network (HAN) to characterize the relationship between network layers. Nevertheless, HAN regarded multiple feature channels of each layer as a whole group and only focused on estimating the correlation between feature channel groups of different layers. Thus, the correlation between the inter-layer feature channels was ignored. Moreover, the 3D convolution in HAN increased the number of parameters and computational burden dramatically.

Distillation Network. The information distillation network is one of the state-of-the-art methods to reduce the number of parameters and achieve a lightweight network architecture. Zheng et al. [36] proposed an information distillation network (IDN) which reduces the computational complexity and memory consumption by channel splitting strategy to downscale the feature maps. Based on IDN, a fast and lightweight information multi-distillation network (IMDN) [37] was presented. IMDN extracted features at a granular level by applying the channel splitting strategy multiple times and proposed contrast-aware channel attention (CCA) to connect the extracted features. However, Liu et al. pointed out that IMDB is still inflexible and inefficient [38]. As a result, they introduced the residual feature distillation block (RFDB), which utilizes the feature distillation connection instead of the channel splitting strategy. Thus, RFDB can improve its performance without introducing additional parameters. While the distillation network has already been employed in some computer vision problems, there are few studies adopt it for image deblurring and deraining. Furthermore, the hierarchical information of different distillation layers is also neglected in the existing methods. Finally, as shown in StyleGAN [39], normalization is very important for low-level vision tasks, but none of the existing distillation methods contains a normalization layer.

## 3. Proposed Method

### 3.1. Overview

The overall architecture of the proposed LFDN is shown in Figure 2, where the green block represents multi-scale hierarchical fusion module (MSHF), red block represents pixel-shuffle up-sampling, and ⊕ represents the operation of elementwise addition. Our LFDN is based on an encoder–decoder structure and consists of two main parts. The feature of blurry or rainy image is firstly extracted by a convolution layer with kernel size as 3 × 3 and step size as 1. Then the extracted feature map is input into the multi-scale hierarchical fusion module (MSHF) in Part I for down-sampling, which acts as the encoder of our model. Through MSHF, the input will be encoded into several small-scale features, and the loss of information will be compensated by the information fusion between different layers. Part II is the multi-feature fusion distillation module (MFFD), which includes several feature distillation normalization blocks (FDNBs) and two attention fusion mechanisms. This part makes the network have the ability to screen useful features by a finer-grained feature extraction strategy while reducing the number of parameters. After part II, it comes to the process of decoder, which combines Residual Convolution block [40] and pixel-shuffle up-sampling block [41] to enlarge the size of feature map to obtain restored image.

### 3.2. Multi-Scale Hierarchical Information Fusion Scheme (MSHF)

As shown in Figure 3, MSHF is a down-sampling module that serves as the encoder in our approach. First, we denote the input feature map as *f*_0_, and its size is W × H × C. Then, the intermediate features *f*_i_ (*i* = 1, 2, 3, 4) of *f*_0_ are progressively extracted using downblock modules [34]. The downblock module consists of two convolution layers, one has a kernel size of 3 × 3, a step size of 2 to make the feature map down-sampled, and the other has a kernel size of 3 × 3, a step size of 1 to make the feature map resampled. Each downblock module [34] will reduce the size of the feature by half and the resampling in it will further refine the feature. After each downblock module for down-sampling, we simply use resblock [40] to realize residual learning. In addition, we fuse different scale features in the small-scale space. The small-scale features after residual learning are up-sampled by pixel-shuffle up-sampling [41] and then elementwise added (denoted by ⊕) with the features of adjacent layers. Finally, the more fine-grained feature on a small scale can be obtained.

Many existing methods [11,12,16,35,36] carried out multiple complex convolution operations at each downsampling layer to prevent the network from losing important detailed information, which will overload the network with large parameters. In our work, we only carry out a small number of feature extraction operations and then fuse multi-scale hierarchical information in small-scale space. This strategy can effectively reduce the computational cost.

The overall algorithm of the proposed MSHF is given in Algorithm 1.
**Algorithm 1** Multi-scale Hierarchical fusion AlgorithmInput: $f_{0}$Output: The output feature of MLHF $f_{M\_out}$1: For *i* in range (1, 5) do2: $f_{i}$ = downblock ($f_{i-1}$)3: $f_{i}$ = resblock ($f_{i-1}$)4: End for5: For *i* in range (3, 5) do6: $f_{i-1}$ = resblock ($f_{i-1}$+upsampling($f_{i}$))7: $f_{i-2}$ = resblock ($f_{i-2}$+upsampling($f_{i-1}$))8: End for9: $f_{M\_out}$ = resblock($f_{2}$)

### 3.3. Multi-Feature Fusion Module Based on Attention Mechanism (MFFD)

#### 3.3.1. Feature Distillation Normalization Block (FDBN)

After MSHF, a MFFD module is proposed to make the network continuously filter the useful channel feature information and eliminate the useless disturbance information. Generally, for reconstructing a sharp image well, it is necessary to increase the number of convolutional layers in the network so that the receptive field can be enlarged to get more information. However, this strategy is not a good choice in practice due to it dramatically increases the number of parameters, which will make the network converge slowly. In order to further reduce the network burden and pursue a lighter and faster network, it is extremely important to extract features through a distillation block. In our network, feature distillation normalization blocks (FDNB) are proposed for the first time in image deblurring and deraining method to extract useful features progressively. As shown in Figure 4, FDNB is mainly composed of the convolution layers with convolution kernel sizes as 1 × 1 and 3 × 3, normalization block, and contrast-aware channel attention layer (CCA-Layer). Before each convolution layer, we perform channel segmentation on the input feature, which divides the feature into two parts. In the first part, the distilled feature $f_{c1\_j}$ can be obtained by the 1 × 1 convolution. The 1 × 1 convolution aims to reduce the number of feature channels and the parameters. Then, we add a normalization layer [39,42] after 1 × 1 convolution. The normalization is designed to ensure the stability of network training and make the network converge more easily. In addition, since normalization can alleviate the dependence of a model on certain dataset, it also helps to improve the generalization ability of our network [43]. In the other part, through the 3 × 3 convolution, the coarse feature $f_{c3\_j}$ is obtained. At the end of FDNB, the CCA-Layer is utilized to concatenate all the distilled features. Specifically, the operation of contrast allows us to obtain feature mappings of multiple spatial vectors. Through a series of FDNBs operations, the performance of our network can be steadily improved.

Overall, we can describe the process of FDNB more clearly with the following Equations:(1)$$f_{c1\_1},\;f_{c3\_1}=F_{s}\left(Conv1\left(f_{in}\right)\right),$$
(2)$$f_{nc1\_1}=Normalization\left(f_{c1\_1}\right),$$
(3)$$f_{c1\_2},\;f_{c3\_2}=F_{s}\left({{Conv}_{2}\left(f\right.}_{c3\_1}+x\right)),$$
(4)$$f_{nc1\_2}=Normalization\left(f_{c1\_2}\right),$$
(5)$$f_{c1\_3},\;f_{c3\_3}=F_{s}\left({{Conv}_{3}\left(f\right.}_{c3\_2}+x\right)),$$
(6)$$f_{nc1\_3}=Normalization\left(f_{c1\_3}\right),$$
(7)$$f_{c3}={Conv}_{4}\left(f_{c3\_3}+x\right),$$
(8)$$f_{out\_i}=f_{in}+CCA\left({Conv}_{5}\left(\left[f_{nc1\_1},f_{nc1\_2},f_{nc1\_3},f_{c3}\right]\right)\right),$$
where *f_in_* is the input feature of FDNB, F_*s*_ denotes the channel splitting operation, Conv${}_{i}$ represents *i*-th convolution layer, *f*${}_{ci\_j}$ denotes the feature obtained by *i* × *i* Conv in *j*-th split operation, *f*${}_{nci\_j}$ represents the feature obtained by *i* × *i* Conv in *j*-th normalization operation, *x* indicates residual. CCA indicates the operation of CCA-Layer, [ ] represents the concatenation operation along the channel dimension, *f*${}_{out\_i}$ is the output of *i*-th FDNB.

#### 3.3.2. Fusion Mechanism

Unlike other networks [36,37,38] that simply stack several distillation modules to hierarchically extract features and only employ final features for the specific task, we rethink the distillation block in our proposed approach. We argue that although the multiple distillation modules (i.e., FDNBs) help the network to obtain richer image information with fewer parameters, the correlation between the intermedia features of each FDNB is ignored. At the same time, we also think about the problem of how to make the information of the last layer fully used. Thus, two different attention based feature fusion mechanisms are proposed to improve the representation ability of extracted features. One is the attention layer fusion module (ALFM) to learn the correlation between feature channels obtained by multiple FDNB layers. The other is the attention channel fusion module (ACFM) to describe the dependency between the inter-channel and intra-channel information in adjacent feature channels of the last layer.

The structure of ALFM is shown in Figure 5. Given the groups of feature channels obtained by *N* FDNB layers with the dimension of N × C × W × H, we first reshape them along the channel and get an NC × WH feature matrix. Then, the feature matrix is transposed and multiplied by itself. In order to normalize the values of attention parameters, the softmax function is utilized in our ALFM. After softmax, we can get an attention matrix to reflect the correlation between channels. At last, we multiply the attention matrix by feature matrix with a scale factor θ and fuse it with the original feature channels by an addition operation. The process can be expressed in Equations (9) and (10):(9)$$m_{ij}=softmax\left(reshape\left(f_{g}\right)⨂{reshape\left(f_{g}\right)}^{T}\right),$$
(10)$$f_{l}=\theta \sum_{i=1}^{N}m_{ij}⊗f_{g}+f_{g},$$
where *f*${}_{g}$ is the input feature channels, ⊗ denotes the matrix multiplication, *m*${}_{ij}$ is the attention matrix, θ is initialized to 0 and it will be automatically optimized by the network. Here, it should be noted that our ALFM treats the feature channels of each FDNB layer separated rather than as a whole. Thus, the correlation between feature channels from both the same and different layers can be considered.

The structure of ACFM is shown in Figure 6. The aim of ACFM is to model the interdependency between feature channels of the last FDNB layer by jointly considering channel and spatial information. Nevertheless, different from Niu et al. [35], who adopted 3D convolution to accomplish this task, our ACFM leverage a Pseudo-3D convolution strategy [44] to reduce the number of parameters. Taking the output of the last FDNB *f*${}_{out\_n}$ as input, we first utilize two convolution kernels with the size of 1 × 3 × 3 and 3 × 1 × 1 to capture the spatial and channel correlations. Then, the attention matrix W obtained after the sigmoid function is element-wise multiplied by *f*${}_{out\_n}$. Finally, the weighted *f*${}_{out\_n}$ is scaled by a factor α and fused with the original features.
(11)$$f_{c}=f_{out\_n}+\alpha \cdot \delta \left(conv2\left(conv1\left(f_{out\_n}\right)\right)\right)⨀f_{out\_n},$$
where δ(·) is the sigmoid function, ⨀ represents element-wise product, conv1 is 1 × 3 × 3 convolution, conv2 is 3 × 1 × 1 convolution, and α is an optimizable parameter which can be optimized by stochastic gradient descent based methods.

Through the attention mechanism in ALFM and ACFM, a more powerful feature representation can be achieved, which will compensate for the potential information loss in lightweight FDNB.

Overall, the loss function of our network can be expressed by Equation (Equation 12):(12)$$L_{MSE}=\frac{1}{wh}\sum_{x=1}^{w}\sum_{y=1}^{h}{\left({\left(I\right)}_{x,y}-LFDN{\left(I_{Blur/Rain}\right)}_{x,y}\right)}^{2},$$
where *I*${}_{Blur/Rain}$ represents the input blurred or rainy image, *LFDN* represents our network. *I* is the standard sharp image, *w* and *h* are the length and width of the input/output image, respectively.

## 4. Experiments

### 4.1. Experimental Settings

We set the number of FDNB layers in our model as 4. The dimension of the feature maps in our fusion module is set as N = 4, C = 50, W = 320, H = 180. For model optimization, we adopt Adam with momentum as 0.9 and weight decay as $10^{-4}$. The learning rate is initialized as $10^{-4}$ and decreases with a factor of 10 for every 5 ×$10^{5}$ iterations. All experiments were conducted using Pytorch on NVIDIA 2080Ti GPUs. Following other image deblurring and deraining works [19,22,45,46], PSNR, SSIM, model size, and inference time are adopted to evaluate our method. For comparative methods in the experiment, their results are directly quoted from their original papers.

#### 4.1.1. Image Deblurring Dataset

Three benchmark datasets are employed for our image deblurring experiment.

GoPro dataset [9] consists of 3214 pairs of blurry and sharp images extracted from 33 sequences captured at 720 × 1280 resolution. The training and testing sets include 2103 and 1111 pairs, respectively.

Kohler dataset [47] consists of 4 images blurred with 12 different kernels for each of them, which is a standard benchmark dataset for evaluating the blind deblurring algorithms.

HIDE dataset [48] has 8422 sharp and blurry image pairs. The images are carefully selected from 31 high-fps videos that contain realistic outdoor scenes containing humans.

#### 4.1.2. Image Deraining Dataset

In our deraining experiment, we use the following synthetic datasets.

The images of synthesizing Rain100L [49] and Rain100H [49] are selected from BSD200 [50]. Rain100L is a synthesized data set with only one type of rain streaks. Rain100H is a synthesized data set with five streak directions.

The Rain14000 [51] collects 1000 clean images from UCID dataset [52], BSD dataset [53], and Google image search to synthesize rainy images. Each clean image was used to generate 14 rainy images with different streak orientations and magnitudes.

The RainTest100 [27] consists of 100 images, where 50 images are randomly chosen from the last 500 images in UCID dataset, and 50 images are randomly chosen from the test set of BSD-500 [54] dataset.

In our deraining experiment, 11,200 images are randomly selected from Rain14000 [51] to form the training set. After training, the remaining 2800 images of Rain14000 [51], Rain100H [49], Rain100L [49], and RainTest100 [54] are used to evaluate the trained network. In image deblurring experiment, we use GoPro [9] dataset that contains 2103 image pairs for training and 1111 pairs for evaluation. Furthermore, to demonstrate generalization performance of our method, we take our GoPro trained model and directly apply it to the test images of HIDE [48] and Kohler datasets [47]. The detailed information of benchmark datasets involved in our experiments can be seen in Table 1.

### 4.2. Quantitative and Qualitative Evaluation on Deraining Task

Following the prior related work [19,22,24,26,27,28,55], we adopt the PSNR and SSIM metrics to evaluate the results of different methods for the image deraining task. It can be clearly seen from Table 2 that the proposed method outperforms other approaches on all four datasets. This indicates our LFDN can effectively remove the image degradation caused by the rain. Meanwhile, the experimental results in Table 2 also mean that our method is not dependent on a specific dataset and has a strong generalization ability to achieve good deraining results in various scenarios. Compared to the most recent MSPFN [19] method, the improvement of our method on Rain14000 [51] is very small, but the performance of our LFDN is much better than MSPFN on the other three benchmark datasets. PreNet [22] and MSPFN [19] also achieve good results on RAIN100L [49] and Rain14000 [51]. However, since there is no feature filter and distillation mechanism in them, their performance is inferior to ours. What is more, thanks to the design of ACFM and ALFM in the proposed LFDN, our method outperforms DerainNet [55]. DIDMDN [24] applied a small receptive field to capture small raindrops with small-scale features, while a large receptive field is employed to capture large raindrops with long scale features. However, due to the lack of fusion mechanism, its performance is worse than some other methods. UMRL [28] and RESCAN [27] use a multi-stage guidance model to generate sharp images, which also results in suboptimal network performance. In addition, in terms of computational complexity, our model runs at least two orders of magnitude faster.

From the qualitative comparison shown in Figure 7, it can be found that the image recovered by DIDMDN [24] is distorted in color, while the image details of UMRL [28] are still insufficient. However, our method can greatly restore the details blocked by the rain. The obvious improvement of our method over some recent works [19,22,24,26,27,28,55] may be attributed to the design of MSHF and MFFD.

### 4.3. Quantitative and Qualitative Evaluation on Deblurring Task

#### 4.3.1. GoPro Dataset

We compare the deblurring performance of our model with some state-of-the-art methods [9,10,11,12,19,48] in terms of PSNR, SSIM, model size, and inference time on GoPro dataset. The quantitative results are shown in Table 3, and visual comparisons are shown in Figure 8. As shown in Table 3, our method owns a smaller model size (i.e., nearly 1.55 M parameters), which is 300X smaller than DeepDeblur [9]. DeepDeblur exploited a multi-scale CNN to restore sharp image, which may cause the heavy parameters. Through introducing the strategies of parameter sharing [10,11,13,20], GAN [15,16], hierarchical multi-patch [12,45], optical flow [21] and motion offsets [46], the number of parameters can be effectively reduced in other comparison methods. Among them, Gao et al. [11] got the smallest number of parameters because it employed a nested skip connection structure for the nonlinear transformation modules to replace stacked convolution layers or residual blocks. However, the authors of reference [11] only focus on the information of the last block and ignore a lot of semantic information contained in the intermedia layers. Different from [11], we utilize the ALFM to obtain the correlation between feature channels from the same and different layers and the ACFM is used to establish the interdependency between feature channels of the last FDNB layer. With the ALFM and ACFM as the fusion methods, the feature information of the middle layer and channels can be reused. Meanwhile, the fusion methods can reduce the weight of the network while ensuring the quality of image repair. Therefore, the PSNR and SSIM obtained by our LFDN are both superior to Gao et al. [11] and other approaches. For more details about the influence of network structure on the performance and model size of our LFDN, please refer to our ablation experiments in Section 4.4. In the visual comparison, it can be seen that most methods cannot recover the sharp object contour, cars, and persons from severe motion blur. For the second and forth images, we can still observe some noticeable blur artifacts in the results of some methods (such as [10,11,12]). Compared with other methods, our model can recover clearer details effectively.

#### 4.3.2. Kohler Dataset

We report the quantitative results on Kohler dataset in Table 4. Visual comparisons are shown in Figure 9. From these results, we could see that our LFDN achieves the best quantitative (PSNR and SSIM) and qualitative performance. Furthermore, please be reminded that similarly to the GoPro case, the model size and average inference time of our method is still less than others.

#### 4.3.3. HIDE Dataset

To verify the validity of our method, we further evaluate our approach on HIDE testing set [48]. From the experimental results in Table 5 and Figure 10, we can clearly find that the performance of our method is better than others and the deblurred images obtained by our LFDN contains more detailed information.

### 4.4. Ablation Study

In this section, we conduct several experiments to evaluate the effectiveness of each component in our method on both deblurring and deraining datasets: GoPro [9] and Rain100H [49]. First, we replace the multi-scale hierarchical fusion module (MSHF) with traditional convolution layers in our network. In this experiment, the stride of convolution in each layer is set as 2, and two residual models are utilized to resampling the feature. Then, the feature distillation normalization blocks (FDNBs) in the decoding stage of our approach are removed and the same number of traditional residual feature distillation blocks (RFDBs) or CNNs are employed for sharp image reconstruction. Finally, we discarded the attention based information fusion modules (i.e., ALFM and ACFM) and employed a standard concatenation to combine the features obtained by different FDNB layers and feature channels of the last FDNB. Through the above settings, we can get five new network structures, i.e., the proposed LFDN without MSHF, without FDNB (replaced by RFDB), without FDNB (replaced by CNN), without ALFM /ACFM and without all of the above components. In order to fairly compare the proposed model with different structures, we adopted the same parameter setting to train the networks. The results of ablation experiments on the GoPro [9] and Rain100H [49] datasets are summarized in Table 6.

As can be seen from Table 6, without any of MSHF, FDNB, or ALFM/ACFM, the image recovery is not optimal. We can find that without MSHF, the value of PSNR decreases by 0.5, and the model size increases by 0.8. This means that MSHF can extract better features with a larger receiver field. Moreover, we can find that performing most of the data computation in the low-scale space is beneficial to reduce the network parameters. The feature fusion in low-scale space is necessary to improve the accuracy. The effectiveness of the FDNB is the most obvious for the network performance. When the FDNB in our model is replaced by RFDB or CNN, its accuracy will be greatly decreased. This is because the RFDB has no ability to normalize the feature maps and CNN cannot filter out useless feature channel information. In order to make the network lightweight without loss of accuracy, we employ ALFM/ACFM mechanism to get finer-grained features. By the ablation experiments, we found that the value of PSNR and SSIM obtained by our model with ALFM/ACFM are higher than without them, which justifies the validity of the proposed ALFM/ACFM mechanism. In addition, we also provide a qualitative visual comparison in Figure 11.

In addition, the sensitivity of our LFDN to the parameters θ and α in ALFM and ACFM modules is tested. As we have mentioned in Section 3.3.2, these two parameters are automatically optimized by our network. Through experiments, we find that the optimal θ obtained by our network are −0.0015 and −0.0008 for deblurring and deraining tasks, while the optimal α obtained by our network are 0.1724 and 0.1137 for deblurring and deraining tasks. In order to justify these optimal parameter values, a comparative experiment is carried out to test the performance of our LFDN under a series of manually set parameter values. From the results in Table 7 and Table 8, it can be found that the parameter values optimized by the network can achieve the best performance on both deblurring and deraining tasks.

Finally, we compare the performances of our LFDN with different numbers of FDNBs. From Table 9, we can find our method performs worse when the number of FDNB layers is small, while too many FDNBs would not further increase its performance. The results also justify that setting the number of FDNB layers as 4 is reasonable in our model.

## 5. Conclusions

In this work, we propose a lightweight fusion distillation network (LFDN) for image deblurring and deraining tasks. In order to make the network have fewer parameters and faster speed, the multi-scale hierarchical fusion module (MSHF) and feature distillation normalization block (FDNB) are adopted in the encoding and decoding stages, respectively. Moreover, two attention based modules are also proposed to improve the feature representation power in our approach. Through a large number of experiments on several benchmark datasets, the following points can be found. Firstly, the MSHF can extract and fuse the image feature in multiple small-scale spaces, which effectively eliminate redundant parameters while maintaining the rich image information. Secondly, the FDNB could help to improve the convergence speed and generalization ability of our proposed model. Thirdly, the attention based ALFM and ACFM modules also contribute to enhancing the performance of our model. Finally, the experimental results also demonstrate that our LFDN outperforms some state-of-the-art methods.

In our future work, we will extend the proposed method to some other image restoration tasks (such as image dehazing and denoising) and compare its performance with related approaches. Furthermore, we will also use other image quality assessment (IQA) methods to evaluate the performance of our method.

## Figures and Tables

**Figure 1 sensors-21-05312-f001:**
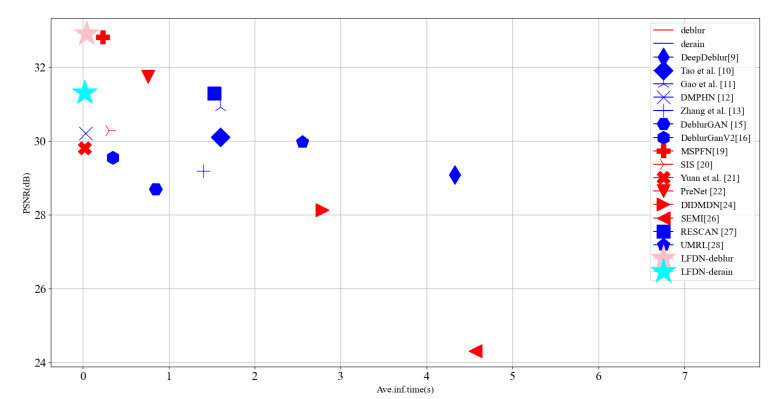
The PSNR and average inference time of state-of-the-art deep learning based deblurring and deraining methods on GoPro and Rain14000 Datasets. Our proposed LFDN achieves the best performance with the least time. (Blue and red marks represent the methods for deblurring and deraining, respectively.)

**Figure 2 sensors-21-05312-f002:**
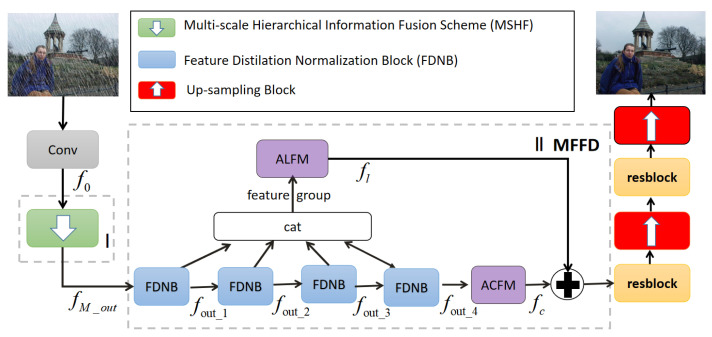
The architecture of lightweight fusion distillation network (LFDN). Part I is the multi-scale hierarchical fusion module (MSHF), Part II is the multi-feature fusion distillation module (MFFD) contains three types of blocks: FDNB is the block of feature distillation normalization, ALFM is the attention layer fusion module, ACFM is the attention channel fusion module. By Part I and Part II, the clear image with more detailed information can be obtained with less network parameters.

**Figure 3 sensors-21-05312-f003:**
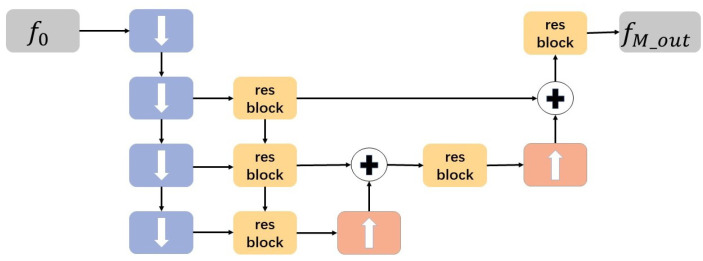
The architecture of the multi-scale hierarchical information fusion scheme (MSHF). The purple block with down arrow represents the downsampling operation, the orange block with up arrow represents the upsampling operation.

**Figure 4 sensors-21-05312-f004:**
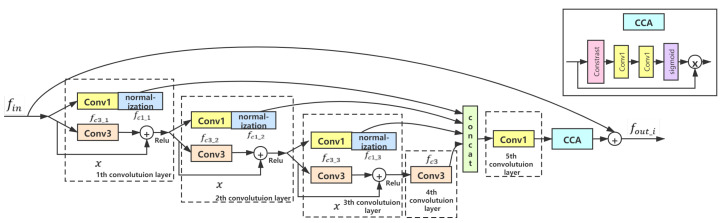
The architecture of the feature distillation normalization block (FDNB), Conv1 represents the convolution kernel sizes as 1 × 1, Conv3 represents the convolution kernel sizes as 3 × 3.

**Figure 5 sensors-21-05312-f005:**
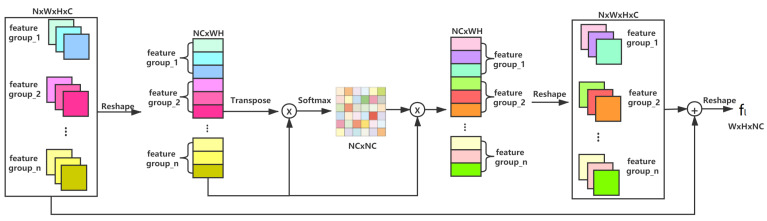
ALFM: The architecture of the attention layer fusion module.

**Figure 6 sensors-21-05312-f006:**
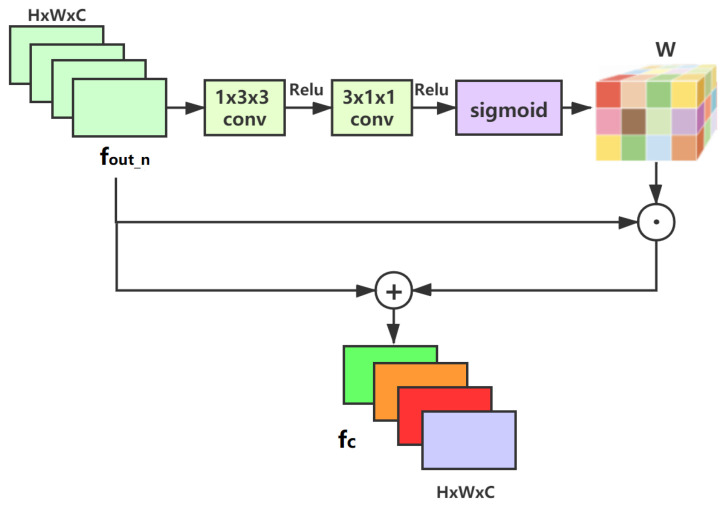
ACFM: The architecture of the attention channel fusion module.

**Figure 7 sensors-21-05312-f007:**
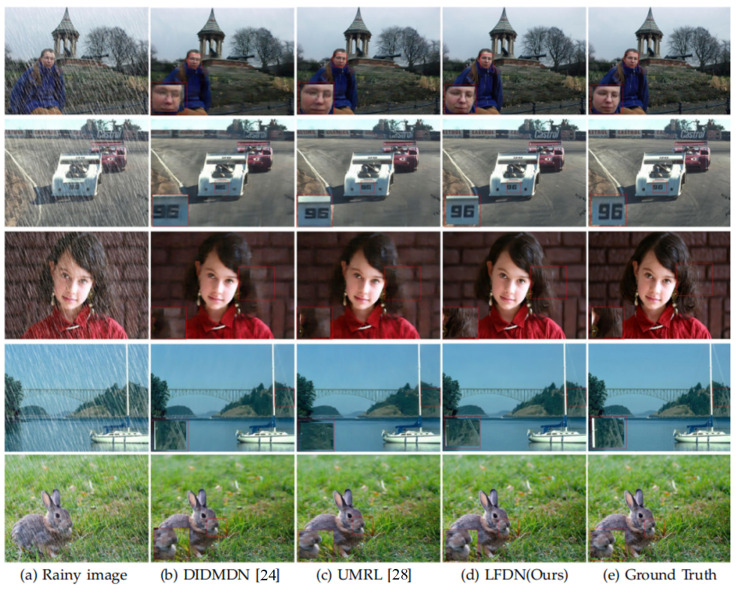
Visual comparison on Rain14000 dataset.

**Figure 8 sensors-21-05312-f008:**
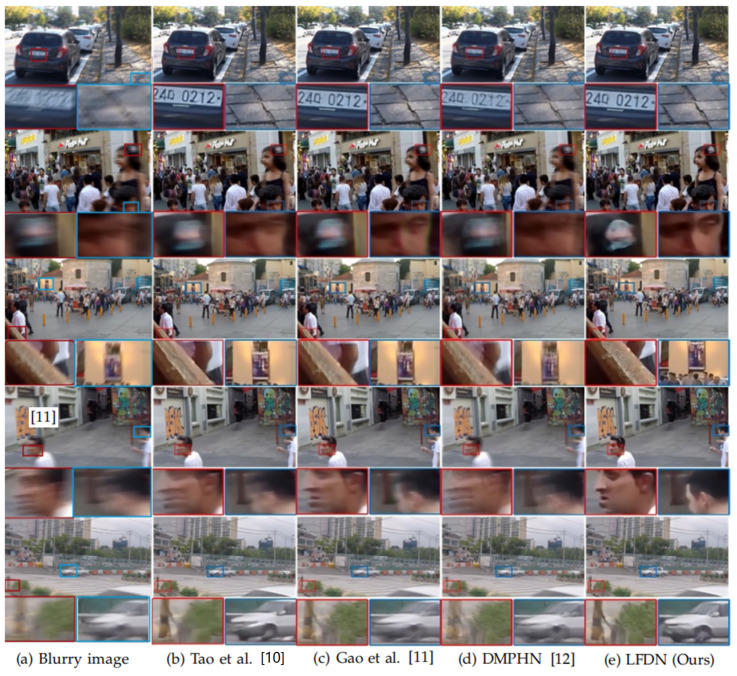
Visual comparison on GoPro dataset.

**Figure 9 sensors-21-05312-f009:**
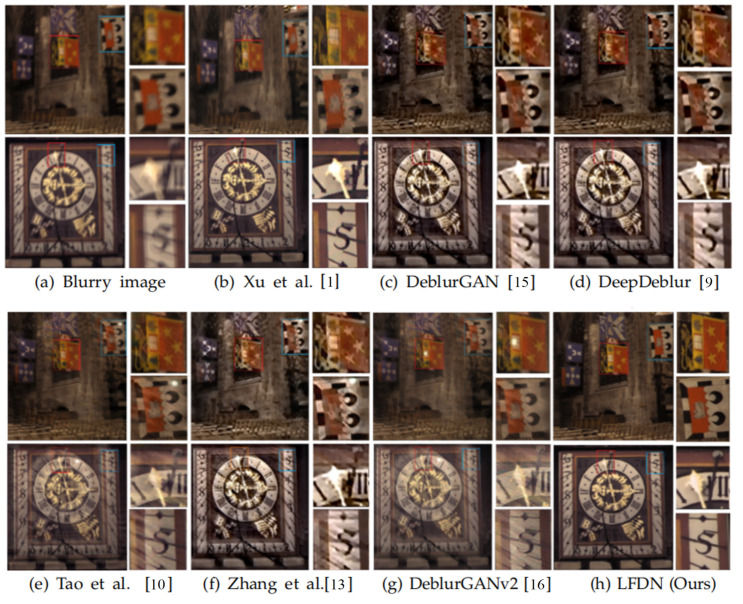
Visual comparison on Kohler dataset.

**Figure 10 sensors-21-05312-f010:**
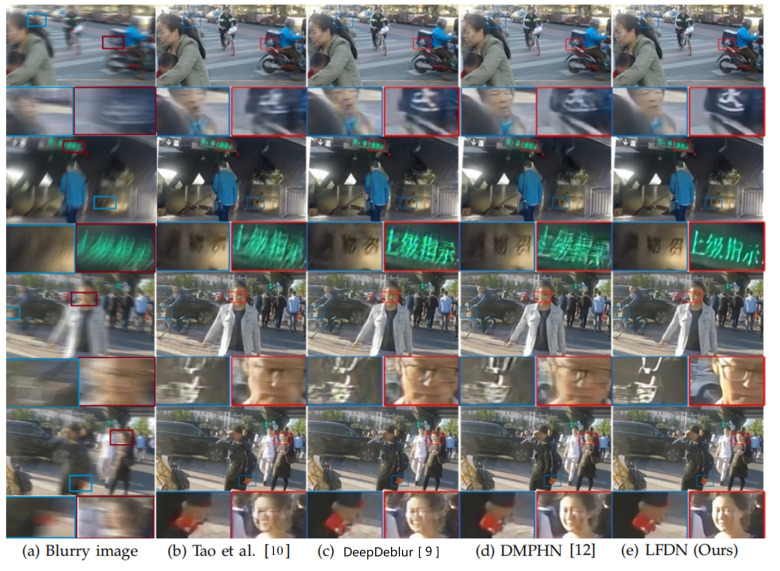
Visual comparison on HIDE dataset.

**Figure 11 sensors-21-05312-f011:**
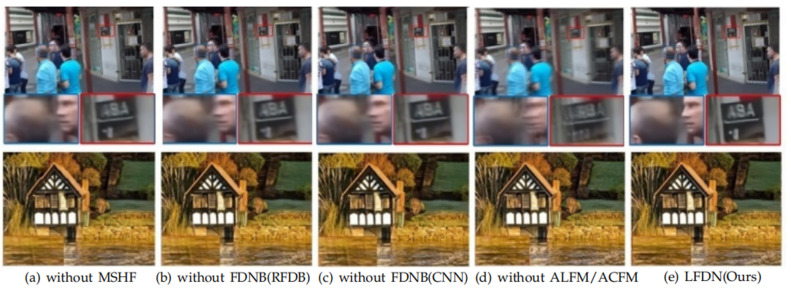
Visual comparison on GoPro (first row) and Rain100H (second row) datasets. (**a**) without MSHF: Only without MSHF. (**b**) without FDNB (RFDB): Use traditional RFDB instead of the FDNB. (**c**) without FDNB (CNN): Use traditional CNN instead of the FDNB. (**d**) without ALFM/ACFM: Only without ALFM/ACFM. (**e**) LFDN: A complete network of ours.

**Table 1 sensors-21-05312-t001:** Dataset description for image deblurring and deraining tasks.

Task	Deraining				Deblurring		
Datasets	Rain14000 [51]	Ranin100L [49]	Rain100H [49]	RainTest100 [54]	GoPro [9]	HIDE [48]	Kolher [47]
Train samples	11,200	0	0	0	2103	0	0
Test samples	2800	100	100	100	1111	2025	64

**Table 2 sensors-21-05312-t002:** Image deraining results on different datasets.

Methods	Rain100H [49]		Rain100L [22]		Rain14000 [51]		Test100 [54]		Ave. Inf. Time (s)
	PSNR	SSIM	PSNR	SSIM	PSNR	SSIM	PSNR	SSIM	
DerainingNet [55]	14.92	0.592	27.03	0.884	24.31	0.961	22.77	0.810	-
SEMI [26]	16.56	0.486	25.03	0.842	24.43	0.782	22.35	0.788	4.567
DIDMDN [24]	17.35	0.524	25.23	0.741	28.13	0.867	22.56	0.818	2.789
UMRL [28]	26.01	0.832	29.18	0.923	29.97	0.923	24.41	0.829	2.552
RESCAN [27]	26.36	0.786	29.80	0.881	31.29	0.904	25.00	0.835	1.530
PreNet [22]	26.77	0.858	32.44	0.950	31.75	0.916	24.81	0.851	0.760
MSPFN [19]	28.66	0.860	32.40	0.933	32.82	0.930	27.50	0.876	0.230
LFDN (Ours)	29.12	0.893	32.79	0.961	32.90	0.935	28.44	0.880	0.043

**Table 3 sensors-21-05312-t003:** Performance and efficiency comparison on GoPro dataset.

Methods	PSNR	SSIM	Model Size (MB)	Ave. Inf. Time (s)
DeepDeblur [9]	29.08	0.841	303.6	15
Zhang et al. [13]	29.19	0.9306	37.1	1.4
Gao et al. [11]	30.92	0.9421	2.84	1.6
DeblurGAN [15]	28.70	0.927	37.1	0.85
Tao et al. [10]	30.10	0.9323	33.6	1.6
DeblurGANv2 [16]	29.55	0.934	15	0.35
DMPHN [12]	30.21	0.9345	21.7	0.03
SIS [20]	30.28	0.912	36.54	0.303
Yuan et al. [21]	29.81	0.9368	3.1	0.01
Tang et al. [45]	31.13	0.9507	31.1	0.088
Zhang et al. [46]	31.05	0.9485	26.3	-
LFDN (Ours)	31.60	0.932	1.55	0.029

**Table 4 sensors-21-05312-t004:** Performance and efficiency comparison on Kohler dataset.

Methods	PSNR	SSIM	Model Size (MB)	Ave. Inf. Time (s)
DeepDeblur [9]	26.48	0.807	303.6	15
Tao et al. [10]	26.57	0.8373	33.6	1.6
Zhang et al. [13]	24.21	0.7562	37.1	1.4
DeblurGAN [15]	26.10	0.807	37.1	0.85
DeblurGANv2 [16]	26.97	0.830	15	0.35
Cai et al. [56]	28.92	0.893	-	1200
Xu et al. [1]	27.47	0.811	-	13.41
LFDN (Ours)	30.98	0.9032	1.55	0.029

**Table 5 sensors-21-05312-t005:** Performance comparison on HIDE dataset.

Methods	Sun et al. [57]	DeepDeblur [9]	Tao et al. [10]	Kupyn et al. [15]	Suin et al. [23]	DMPHN [12]	LFDN (Ours)
PSNR	23.21	27.43	28.60	26.44	29.98	29.09	30.07
SSIM	0.797	0.902	0.928	0.890	0.930	0.924	0.932
modle size (MB)	-	303.6	33.6	37.1	-	86.8	1.55
Ave. Inf. Time (s)	23.45	4.33	1.6	0.85	0.77	0.98	0.029

**Table 6 sensors-21-05312-t006:** Ablation experimental results of our network on GroPro and Rain100H datasets.

MSHF	FDNB	ALFM/ACFM	PSNR on GroPro	SSIM on GroPro	PSNR on Rain100H	SSIM on Rain100H	Model Size (MB)
X	X	X	28.71	0.901	27.36	0.862	33.6
X	✓	✓	30.84	0.917	28.92	0.885	1.8
✓	X (RFDB)	✓	29.28	0.891	28.48	0.871	2.9
✓	X (CNN)	✓	28.56	0.882	28.04	0.867	4.3
✓	✓	X	31.01	0.921	28.98	0.886	2.4
✓	✓	✓	31.60	0.932	29.12	0.893	1.55

**Table 7 sensors-21-05312-t007:** The performance comparison of different θ values.

θ	0.002	0.001	0	−0.001	−0.002	−0.0015	−0.0008
PSNR on GoPro	31.48	31.51	31.51	31.56	31.55	31.60	31.23
PSNR on Rain100H	28.95	28.97	29.01	29.10	29.07	29.01	29.12

**Table 8 sensors-21-05312-t008:** The performance comparison of different α values.

α	0.3	0.2	0.1	0	0.1724	0.1137
PSNR on GoPro	31.37	31.55	31.48	31.25	31.60	31.53
PSNR on Rain100H	28.44	29.02	29.09	28.99	29.03	29.12

**Table 9 sensors-21-05312-t009:** Performance comparison of our method under different numbers of distillation blocks.

Distillation Blocks Setting	1	2	3	4	5	6
PSNR on GroPro	29.82	29.73	30.77	31.60	31.52	31.61
PSNR on Rain100H	28.03	28.14	28.88	29.12	29.11	29.13
Model Size (MB)	1.46	1.49	1.52	1.55	1.58	1.61

## Data Availability

The code of LFDN can be available at https://github.com/LiQiang0307/LFDN (accessed on 5 August 2021).

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
