# Peer review of "A Lightweight Fusion Distillation Network for Image Deblurring and Deraining [Author-notes fn1-sensors-21-05312]"

_sensors, 2021, doi:10.3390/s21165312_

Round 1
Reviewer 1 Report
This paper proposes a novel lightweight fusion distillation network (LFDN) with an encoder-decoder architecture to distill the useful features and eliminate redundant parameters. A multi-scale hierarchical information fusion scheme (MSHF) is proposed to encode the image with rain and blur, which extracts and fuses the image feature in multiple small-scale spaces, and eliminates redundant parameters. In the decoding stage, a feature distillation normalization block is designed to enable the network to distill and screen valuable channel information of feature maps continuously. Finally, two attention mechanism based modules are also presented in the decoding process to exploit the interdependency between the layers and feature channels. The proposed method is simple and effective, and redundant parameters can be eliminated significantly by the proposed MSHF while maintaining the rich image information. In my opinion, the article has relatively strong originality, and meets the requirements of publication. I recommend accepting it but with some modifications.
- In this paper, how does the author get the setting of the factors such as α and θ? It is recommended that the author conduct a comparative test of the parameters to enrich the content.
- The description of Algorithm 1 is not elegant enough, it is recommended to align it to the left to enhance readability. Moreover the layout of Table 1 overflows the border, which may appear that the author is not professional enough.
- It is strongly recommended that the author add relevant logical descriptions under the figures to make the article more readable.
- Different font formats are used at the bottom of Figures 7, 8, 9, 10. The author is strongly recommended to check similar problems in detail to make the article more descriptive.
- Many methods and experimental parts are very descriptive, and the comparison methods of the last 2 years are expected to be compared in the article.
- Missing articles and other incorrect English language constructs are distracting and make interpretation difficult. The authors are advised to correct the linguistic errors in the edited text.
- The authors ignore some relevant papers. For example, single-image dehazing methods that have been published in 2020 and 2021 please see (“Image Dehazing by an Artificial Image Fusion Method Based on Adaptive Structure Decomposition,” IEEE Sensors Journal, vol.20, and “A Novel Fast Single Image Dehazing Algorithm Based on Artificial Multiexposure Image Fusion,” IEEE Transactions on Instrumentation and Measurement, vol.70). The authors should compare their method with it carefully.
Author Response
- Comment: In this paper, how does the author get the setting of the factors such as α and θ? It is recommended that the author conduct a comparative test of the parameters to enrich the content.
Response: In our proposed LFDN, the parameters α and θ in ALFM and ACFM are optimized by the network. According to your suggestion, we have further clarified this point in Section 3.2.2 (please see our analysis under Equations 10 and 11). Furthermore, a comparison experiment is also carried out to test the performance of our model under various α and θ values in Tables 7 and 8. From the experimental results in these tables, it can be found that parameter values optimized by the network can achieve the best performance on both deblurring and deraining tasks.
- Comment: The description of Algorithm 1 is not elegant enough, it is recommended to align it to the left to enhance readability. Moreover the layout of Table 1 overflows the border, which may appear that the author is not professional enough.
Response: We have reformatted Algorithm 1 and Table 1 according to your suggestion.
- Comment: It is strongly recommended that the author add relevant logical descriptions under the figures to make the article more readable.
Response: According to your suggestion, we have added some necessary descriptions below the key figures (such as Figures 1-4) to make the paper more readable.
- Comment: Different font formats are used at the bottom of Figures 7, 8, 9, 10. The author is strongly recommended to check similar problems in detail to make the article more descriptive.
Response: We are very sorry for our negligence of using different font formats at the bottom of Figure 7,8,9, and 10. During the revision, we have carefully checked the font formats throughout the manuscript and avoid similar problems.
- Comment: Many methods and experimental parts are very descriptive, and the comparison methods of the last 2 years are expected to be compared in the article.
Response: According to your suggestion, the following new methods proposed in the last two years have been compared in our experiment.
Tang, K.; Xu, D.; Liu, H.; Zeng, Z. Context Module Based Multi-patch Hierarchical Network for Motion Deblurring. Neural Process. Lett. 2021, 53, 211–226.
Zhang, Y.; Wang, C.; Maybank, S.J.; Tao, D. Self-supervised Exposure Trajectory Recovery for Dynamic Blur Estimation. CoRR 2020, abs/2010.02484, [2010.02484].
Suin, M.; Purohit, K.; Rajagopalan, A.N. Spatially-Attentive Patch-Hierarchical Network for Adaptive Motion Deblurring. 2020 IEEE/CVF Conference on Computer Vision and Pattern Recognition, CVPR 2020, Seattle, WA, USA, June 13-19, 2020. IEEE, 2020, pp. 3603–3612.
Zhang, H.; Dai, Y.; Li, H.; Koniusz, P. Deep Stacked Hierarchical Multi-Patch Network for Image Deblurring. IEEE Conference on Computer Vision and Pattern Recognition, CVPR 2019, Long Beach, CA, USA, June 16-20, 2019. Computer Vision Foundation /IEEE, 2019, pp. 5978–5986.
- Comment: The authors ignore some relevant papers. For example, single-image dehazing methods that have been published in 2020 and 2021 please see (“Image Dehazing by an Artificial Image Fusion Method Based on Adaptive Structure Decomposition,” IEEE Sensors Journal, vol.20, and “A Novel Fast Single Image Dehazing Algorithm Based on Artificial Multiexposure Image Fusion,” IEEE Transactions on Instrumentation and Measurement, vol.70). The authors should compare their method with it carefully.
Response: According to your suggestion, we have cited these two references in our revised manuscript (please see Ref. [8] and [9]) and analyzed the difference between them and the deep learning based methods (including our proposed method since it is also based on deep learning). However, since these two papers are both proposed for image dehazing rather than deblurring or deraining, and the authors of these two papers did not provide the resource codes, we cannot know their performance for image deblurring and deraining tasks, nor implement the methods proposed in these two papers on our datasets. Therefore, the deblurring and deraining results of these two methods are not compared in our experiment. In our future work, we will extend our method to other image restoration tasks (such as dehazing and denoising) and compare its performance with the methods in [8] and [9].
Reviewer 2 Report
In the paper, an approach to image deblurring and deraining is proposed. Comments:
1. It is claimed that the proposed module “filters out useless feature channel information”. The claim should be supported experimentally using a convincing example.
2. The connection of the paper with the journal is not shown. Are there similar works here? Why is it suitable?
3. The performance is assessed using image quality assessment (IQA) methods: SSIM and PSNR. They are often used as borderline perceptual models in recent IQA works. Here, please use a more recent method such as VSI or one of the deep learning IQA approaches with available source code.
4. It is mentioned that in the work of Gao et al. [10] a lot of semantic information is ignored. Does the proposed approach make the full usage of such information? Please elaborate.
5. Section 5 is quite superficial. Please summarize the main findings and contributions here. Now, the abstract is more informative.
6. The results cannot be replicated. Please share the source code with readers, ensuring the repeatability of the results.
Author Response
1.Comment: It is claimed that the proposed module “filters out useless feature channel information”. The claim should be supported experimentally using a convincing example.
Response: We are very sorry for neglecting to use experiments to prove that our FDNB module can filter out useless feature channel information. In our revised manuscript, we have added an ablation experiment in which the FDNB module in our network is replaced by standard CNN (please see Table 6). From the experimental result, we find that since CNN does not have the ability for feature channel filtering and treat all feature channels equally, its performance is inferior to the proposed FDNB. Moreover, from the image deblurring result in Figure 11, we can also see that replacing FDNB by CNN cannot restore the sharp image well.
We think the ablation experiment and Figure 11 can be convincing examples to support that our proposed FDNB nodule can filter out useless feature channel information.
2.Comment: The connection of the paper with the journal is not shown. Are there similar works here? Why is it suitable?
Response: As we have mentioned in Section 1, both the blur and rain streaks will degrade the image captured by the vision sensors and people’s visual perception of detailed image information. Since most imaging sensors and devices are not robust or efficient enough to deal with degradation caused by blur and rain, it is necessary to develop an effective model to mitigate adverse blur and rain effects to obtain high-visibility images, so that many computer vision tasks can be well performed.
According to the above analysis, we think our paper is related to this journal. Furthermore, we did a keyword (image deblurring) search in this journal and find there are many similar works. For example:
Zhou, F.; Yang, J.; Jia, L.; Yang, X.; Xing, M. Ultra-High Resolution Imaging Method for Distributed Small Satellite Spotlight MIMO-SAR Based on Sub-Aperture Image Fusion. Sensors 2021, 21(5), 1609; https://doi.org/10.3390/s21051609.
https://www.mdpi.com/1424-8220/21/5/1609
Wu, X.; Li, J.; Zhou, G.; Lü, B.; Li, Q.; Yang, H. RRG-GAN Restoring Network for Simple Lens Imaging System. Sensors 2021, 21(10), 3317; https://doi.org/10.3390/s21103317.
https://www.mdpi.com/1424-8220/21/10/3317
Sun, S.; Duan, L.; Xu, Z.; Zhang, J. Blind Deblurring Based on Sigmoid Function. Sensors 2021, 21(10), 3484; https://doi.org/10.3390/s21103484.
https://www.mdpi.com/1424-8220/21/10/3484
Feng, H.; Guo, J.; Xu, H.; Ge, S. SharpGAN: Dynamic Scene Deblurring Method for Smart Ship Based on Receptive Field Block and Generative Adversarial Networks. Sensors 2021, 21(11), 3641; https://doi.org/10.3390/s21113641.
https://www.mdpi.com/1424-8220/21/11/3641
3.Comment: The performance is assessed using image quality assessment (IQA) methods: SSIM and PSNR. They are often used as borderline perceptual models in recent IQA works. Here, please use a more recent method such as VSI or one of the deep learning IQA approaches with available source code.
Response: In the research field of image deblurring and deraining, SSIM and PSNR are two widely used measurements to evaluate the performance of algorithms and have been adopted by many state-of-the-art works. In our paper, we need to compare the performance of the proposed method with these state-of-the-art works to demonstrate its effectiveness. In the experiment, the results of all comparison methods are directly quoted from their original papers. Therefore, since these SOTA methods all adopted SSIM and PSNR for performance evaluation and did not provide results of other measurement in their papers, we can only use the same measurements as them for comparison.
Although we could implement the codes of comparison methods by ourselves to obtain the image deblurring and deraining results and then use other IQA methods for performance assessment, it will take a very long time. Besides, some of the comparison methods do not provide their source code. Thus, we will use other IQA methods to evaluate the performance of our method in the future work (please see Section 5 of our revised paper).
4.Comment: It is mentioned that in the work of Gao et al. [10] a lot of semantic information is ignored. Does the proposed approach make the full usage of such information? Please elaborate.
Response: According to your suggestion, we have added detailed analysis to compare our method with Gao et al. [12] in Section 4.3.1 of the revised paper. Different from [12], we utilize the ALFM to obtain the correlation between feature channels from the same and different layers and the ACFM is used to establish the interdependency between feature channels of the last FDNB layer. With the ALFM and ACFM as the fusion methods, the features information of the middle layer and channels can be reused. Meanwhile, the fusion methods can reduce the weight of the network while ensuring the quality of image repair. Therefore, the PSNR and SSIM obtained by our LFDN are both superior to Gao et al. [12] and other approaches.
5.Comment: Section 5 is quite superficial. Please summarize the main findings and contributions here. Now, the abstract is more informative.
Response: We have re-written Section 5 to summarize the main findings and contributions in our paper. Moreover, some future work has also been provided in this section.
6.Comment: The results cannot be replicated. Please share the source code with readers, ensuring the repeatability of the results.
Response: Considering this suggestion, our code has been uploaded to github and can be available at: https://github.com/LiQiang0307/LFDN.
Round 2
Reviewer 1 Report
I have carefully checked this manuscript, I suggest to accept this paper.
Author Response
We appreciate your careful review and constructive comment to improve our paper.
Reviewer 2 Report
Authors have addressed my comments. I'm glad they decided to publish their source code.
Author Response

(The authors gave the same response as above.)
